# Differential Regulation of Intracisternally Injected Angiotensin II-Induced Mechanical Allodynia and Thermal Hyperalgesia in Rats

**DOI:** 10.3390/biomedicines11123279

**Published:** 2023-12-12

**Authors:** Ki-Don Park, Jo-Young Son, Hak-Kyun Kim, Yu-Mi Kim, Jin-Sook Ju, Min-Jeong Jo, Min-Kyoung Park, Min-Kyung Lee, Dong-Kuk Ahn

**Affiliations:** 1Department of Oral Physiology, School of Dentistry, Kyungpook National University, Daegu 41940, Republic of Korea; turtles1@hanmail.net (K.-D.P.); n-violetjy@nate.com (J.-Y.S.); hakyuna@hanmail.net (H.-K.K.); boboatom@naver.com (Y.-M.K.); jsju@knu.ac.kr (J.-S.J.); dmvvv3@naver.com (M.-J.J.); 2Department of Dental Hygiene, Kyung-Woon University, Gumi 39160, Republic of Korea; bukimin@hanmail.net; 3Department of Dental Hygiene, Dong-Eui University, Busan 47340, Republic of Korea; lmk849@deu.ac.kr

**Keywords:** angiotensin II, angiotensin II type 1 receptor, angiotensin II type 2 receptor, astrocytes, botulinum toxin type A, mechanical allodynia, thermal hyperalgesia

## Abstract

The present study examined the underlying mechanisms of mechanical allodynia and thermal hyperalgesia induced by the intracisternal injection of angiotensin (Ang) II. Intracisternal Ang II injection decreased the air puff threshold and head withdrawal latency. To determine the operative receptors for each distinct type of pain behavior, we intracisternally injected Ang II receptor antagonists 2 h after Ang II injection. Losartan, an Ang II type 1 receptor (AT1R) antagonist, alleviated mechanical allodynia. Conversely, PD123319, an Ang II type 1 receptor (AT2R) antagonist, blocked only thermal hyperalgesia. Immunofluorescence analyses revealed the co-localization of AT1R with the astrocyte marker GFAP in the trigeminal subnucleus caudalis and co-localization of AT2R with CGRP-positive neurons in the trigeminal ganglion. Intracisternal pretreatment with minocycline, a microglial inhibitor, did not affect Ang II-induced mechanical allodynia, whereas L-α-aminoadipate, an astrocyte inhibitor, significantly inhibited Ang II-induced mechanical allodynia. Furthermore, subcutaneous pretreatment with botulinum toxin type A significantly alleviated Ang II-induced thermal hyperalgesia, but not Ang II-induced mechanical allodynia. These results indicate that central Ang II-induced nociception is differentially regulated by AT1R and AT2R. Thus, distinct therapeutic targets must be regulated to overcome pain symptoms caused by multiple underlying mechanisms.

## 1. Introduction

Angiotensin (Ang), a peptide hormone, is formed by renin from its substrate, angiotensinogen, in a coordinated hormonal cascade of the renin angiotensin system (RAS). Ang II, a major bioactive peptide, regulates vascular contraction, hormone secretion, and fluid balance [1,2]. However, peripheral Ang II is inactive in the central nervous system as the blood–brain barrier is impermeable to peripheral RAS-related components. Therefore, Ang II exerts localized active effects on receptors in the central nervous system. Previous studies have shown that angiotensinogen is found in the cerebrospinal fluid [3] and the brain originates from astroglial cells [4,5,6]. Furthermore, the angiotensin-converting enzyme (ACE) is prominently expressed throughout the brain [7]. These results indicate that Ang II has important functions in the central nervous system in addition to its peripheral effects.

Emerging findings suggest that central Ang II contributes to the modulation of nociceptive progression in the central nervous system. The intrathecal administration of Ang II induces nociceptive scratching behavior mediated by p38 MAPK signaling [8]. In addition, the intrathecal administration of losartan, an angiotensin II type 1 receptor (AT1R) antagonist, which inhibits p38 MAPK phosphorylation, exerts significant antinociceptive effects in the mouse formalin test regarding pain [9]. These results indicate that Ang II plays an important role in nociceptive processing at the spinal cord level.

Ang II-related substances were recently developed and reported as painkillers. The analgesic efficacy of EMA401, a highly selective angiotensin II type 2 receptor (AT2R) antagonist, in patients with postherpetic neuralgia and painful diabetic neuropathy was reported in randomized, double-blind clinical studies [10,11]. However, EMA 401 did not penetrate the central nervous system [12].

In addition to the effects at the spinal cord level, radioimmunological and immunocytochemical studies showed the location of Ang II, which colocalizes with substance P, in trigeminal ganglions to innervate the orofacial area in rats and humans [13]. Although these data suggest the involvement of Ang II in nociceptive processing in the orofacial area, the underlying mechanisms of central Ang II in the modulation of nociceptive transmission in the orofacial area remain unclear.

The present study investigated mechanical allodynia and thermal hyperalgesia in the orofacial area after intracisternal Ang II injection. The receptors responsible for the underlying mechanism of the induction of nociceptive behavior by intracisternal Ang II injection were also identified via immunohistochemical and behavioral analyses.

## 2. Materials and Methods

### 2.1. Animals

Male Sprague Dawley rats weighing approximately 220–240 g were used in the experiments. The rats were placed in a room with a constant temperature and a 12 h/12 h light/dark cycle. Food and water were provided ad libitum. All experiments were conducted by investigators who were blinded to the study. The procedures were approved by the Laboratory Animal Care and Use Committee of the Kyungpook National University (degree E, KNU 2023-0050). Furthermore, the experiments followed the ethical guidelines for the experimental investigation of pain in conscious animals set forth by the International Association for the Study of Pain.

### 2.2. Intracisternal Catheterization

The experimental animals were anesthetized using a mixed solution of ketamine (40 mg/kg; intramuscular) and xylazine (4 mg/kg; intramuscular). The anesthetized rats were fixed in a stereotaxic frame, as previously described [14,15] and then implanted with polyethylene tubing (PE10; Clay Adams, Parsippany, NJ, USA). The end of the PE tube was inserted into a very small hole in the atlantooccipital membrane and dura mater. The cannula tip of the PE 10 tube was positioned at the obex level of the medulla. The other end of the tube was guided subcutaneously to the top of the skull and secured with stainless-steel screws and dental resin. Animals with impaired motor function or catheter malposition following intracisternal catheter placement were excluded from further analysis. Animals were allowed to recover for 72 h postoperatively as previously described [16,17,18], and then chemicals were injected intracisternally after the recovery period.

### 2.3. General Procedures for Behavioral Testing

For behavioral observations, each rat was placed in a custom-made cylindrical restraint measuring 40–60 and 70–120 mm in length and length, respectively. The animals were acclimatized to the experimental environment for at least 30 min before commencement of the experiment. All behavioral tests were conducted between 09:00 and 18:00 h.

### 2.4. Evaluation of Mechanical Allodynia 

Withdrawal behavior, including aggressive biting and escape responses from air-puff stimulation, was measured after air-puff stimulation was performed 10 times successively, lasting 4 s at 10 s intervals, as previously described [19,20,21]. Air-puff pressure (intensity, duration, and intervals) was adjusted using a pico-injector (Harvard Apparatus, Holliston, MA, USA). Air-puff stimulation was applied using a metal tube (26 gauge, 10 cm long) positioned 1 cm away from the facial skin at a 90° angle at the most sensitive area among the lower jaw of the facial region, mouth angle, and vibrisa pad area. Mechanical allodynia was assessed when the experimental animal exhibited a nociceptive behavioral response in 50% of air-puff stimulations. The cutoff pressure was 40 psi, as previously described [22,23,24]. Naïve rats showed no escape response to pressures below 40 psi.

### 2.5. Evaluation of Orofacial Heat Hyperalgesia 

Heat hyperalgesia was assessed by applying radiant heat to the skin and then measuring head withdrawal latency, as previously described. [25]. Noxious heat stimulation was applied using an infrared thermal stimulator (Infrared Diode Laser, LVI-808-10, LVI tech, Seoul, Republic of Korea) with the stimulation intensity set to 11 W and 18.1 A. The intensity of the thermal stimulus that produced a constant head-withdrawal latency (approximately 12 s) when applied at a 10 cm distance from the heat source was measured. Thermal stimulation was performed twice, with at least 3 min between stimulations. A cut-off time of 20 s was set as the threshold to avoid potential tissue damage.

### 2.6. Immunofluorescence Staining

Under anesthesia, the rats (*n* = 6/group) were perfused with a freshly prepared solution of 4% paraformaldehyde in 0.1 M phosphate buffer (PB, pH 7.4). The caudal medulla and trigeminal ganglion were extracted, and the extracted tissues were post-fixed in 4% paraformaldehyde for 2 h at 4 °C and cryoprotected with 30% sucrose in 0.1 M PB overnight. The tissues were cut on a freezing microtome at 16 μm and processed for immunofluorescence.

For double immunofluorescence staining, sections from the caudal medulla and trigeminal ganglion were treated with 50% ethanol for 30 min and then blocked with 10% normal donkey serum (Jackson ImmunoResearch, West Groove, PA, USA) in phosphate buffered saline (PBS, pH 7.4) for 1 h at room temperature. After rinsing with 0.1 M PBS, the sections were incubated overnight at 4 °C with rabbit AT1R (1:10, Proteintech, Chicago, IL, USA) mixed with mouse anti-NeuN (neuronal marker; 1:1000; Millipore) (Burlington, MA, USA), goat anti-Iba1 (microglial marker; 1:1000; Millipore), or mouse anti-GFAP (astrocyte marker; 1:1000; Millipore) and a rabbit AT2R (1:100, Huabio, Woburn, MA, USA) mixed with mouse anti-CGRP(c-terminal marker; 1:1000; Abcam) (Cambridge, MA, USA). On the next day, the sections were incubated with a mixture of anti-rabbit Cy3 and anti-goat or anti-mouse FITC antibodies (1:200; Jackson ImmunoResearch). The sections were mounted on slides using Vectashield (Vector Laboratories, Burlingame, CA, USA). Stained sections were examined under a fluorescence microscope (BX 41 and U-RFL-T; Olympus, Tokyo, Japan).

### 2.7. Chemicals

Ang II, minocycline and L-α-aminoadipate (LAA) were bought from Sigma Aldrich (St. Louis, MO, USA). Losartan and PD123319 were purchased from Tocris (Bristol, UK). All chemicals were dissolved in saline. Botulinum toxin type A (BTX-A, Botulax®, 100 units) diluted in 1 mL saline, was donated by Hugel Inc. (Chuncheon, Republic of Korea).

### 2.8. Experimental Protocols

#### 2.8.1. Pronociceptive Effects of Intracisternally Administered Ang II

Ang II (0.1, 0.5, or 2.5 ng in 10 μL) was intracisternally administered to naïve rats (*n* = 7/group). To evaluate mechanical allodynia and thermal hyperalgesia, changes in air-puff thresholds and head withdrawal latency were measured 0.5, 1, 2, 3, 4, 6, 24, and 48 h after intracisternal injection of Ang II or vehicle.

#### 2.8.2. Effects of Ang II Receptor Blockade on Ang II-Induced Mechanical Allodynia and Thermal Hyperalgesia

To identify which Ang II receptors regulated Ang II-induced mechanical allodynia or thermal hyperalgesia in rats, a AT1R antagonist (losartan, 10 or 20 μg/10 μL) or a AT2R antagonist (PD123319, 150 or 300 μg/10 μL) was intracisternally injected 2 h after Ang II injection (*n* = 7/group). Changes in air-puff thresholds were measured 0.16, 0.5, 1, 1.5, 2, 3, 5, and 24 h and changes in head withdrawal latency were also measured 0.5, 1, 1.5, 2, 3, 5, and 24 h after the administration of AT1R and AT2R antagonists or vehicle.

#### 2.8.3. Colocalization of Ang II Receptors in Naïve Rats

Double immunostaining was performed for AT1R, which colocalizes with GFAP (an astrocyte marker), Iba1(a microglial marker), or NeuN (a neuronal marker) in the trigeminal subnucleus caudalis (*n* = 6/group). Furthermore, double immunostaining was carried out for AT2R, which colocalizes with CGRP in the trigeminal subnucleus caudalis and the trigeminal ganglion.

#### 2.8.4. Role of Glial Cells in Ang II-Induced Mechanical Allodynia

To investigate the role of glial cells, changes in air-puff thresholds after the blockade of glial cell activity in Ang II-treated rats were measured. To block microglial activity, minocycline (50 or 100 μg) was intracisternally injected 10 min before Ang II injection. Moreover, LAA (4 or 8 μg), an astrocyte inhibitor, was intracisternally injected 10 min before Ang II injection. Changes in air-puff thresholds were measured 0.5, 1, 2, 3, 4, 6, and 24 h following intracisternal Ang II injection (*n* = 7/group). Vehicle was used as a control for glial inhibitors.

#### 2.8.5. Effects of BTX-A on Ang II-Induced Thermal Hyperalgesia

To determine whether the primary afferent nerve is involved in Ang II-induced thermal hyperalgesia, changes in head withdrawal latency were measured in rats subcutaneously injected with BTX-A (3 U/kg) or vehicle in the vibrissa pad 3 days before Ang II injection. This timepoint was chosen based on previous findings that BTX-A exerts significant anti-allodynic effects 3 days after injection [26] Changes in head-withdrawal latency were examined 0.5, 1, 2, 3, 4, 6, and 24 h after intracisternal Ang II injection (*n* = 7/group).

### 2.9. Data Analysis

Differences in nociceptive behavior between groups were examined via repeated-measures analysis of variance followed by Holm Sidak post-hoc analysis. For all comparisons, *p* < 0.05 was considered to indicate statistical significance. All data were expressed as mean values ± SEM.

## 3. Results

### 3.1. Intracisternal Administration of Ang II Induces Nociceptive Behavior

Figure 1 presents changes in air-puff thresholds and head-withdrawal latency after intracisternal Ang II injection. Following intracisternal Ang II injection, mechanical allodynia and thermal hyperalgesia were observed in the trigeminal region, including the upper and lower jaws, the mouth angle area, and vibrisa pad. The intracisternal administration of Ang II (0.1, 0.5, or 2.5 ng) significantly decreased air-puff thresholds (F_(3,24)_ = 362.1, *p* < 0.05) in a dose-dependent manner. Decrements in the air-puff threshold occurred 2 h post-injection (2.5 ng Ang II) and was maintained for 24 h post-injection. However, intracisternally injected vehicles exerted no effect on air-puff thresholds. Intracisternal Ang II injection also caused thermal hyperalgesia. A high dose of Ang II (2.5 ng) decreased the latency of head withdrawal from thermal stimulation (F_(3,24)_ = 89.99, *p* < 0.05). Thermal hyperalgesia occurred 1 h post-injection and returned to basal values 6 h post-injection. Neither low-dose Ang II (0.1 or 0.5 ng) nor the vehicle affected the latency of head withdrawal.

### 3.2. Contribution of Ang II Receptors to Ang II-Induced Nociceptive Behavior

Figure 2 presents which Ang II receptors were responsible for Ang II-induced mechanical allodynia and thermal hyperalgesia. The intracisternal administration of 20 μg losartan, an AT1R antagonist, significantly attenuated mechanical allodynia (F_(2,18)_ = 89.99, *p* < 0.05, Figure 2A). Antiallodynic effects occurred 1 h post-injection, were maintained for 5 h, and had recovered to basal values 24 h post-injection of the AT1R antagonist. Neither a low-dose losartan (10 μg) nor the vehicle affected the air-puff thresholds. Losartan injection did not affect Ang II-induced thermal hyperalgesia (Figure 2B).

The present study also examined changes in Ang II-induced mechanical allodynia and thermal hyperalgesia after intracisternal treatment with PD123319, an AT2R antagonist. Intracisternal administration of PD123319 did not affect Ang II-induced mechanical allodynia (Figure 2C). However, 150 and 300 μg PD123319 significantly inhibited thermal hyperalgesia compared with the vehicle (F_(2,18)_ = 10.52, *p* < 0.05, Figure 2D).

### 3.3. Colocalization of Ang Receptor in Naïve Rats

Figure 3 illustrates double immunofluorescence staining for AT1R and AT2Rs with cell-specific markers to determine where the receptors were expressed in the ipsilateral trigeminal subnucleus caudalis and the trigeminal ganglion. AT1R colocalized with the astrocyte marker (GFAP) (Figure 3A), whereas AT2R colocalized with CGRP in the terminal of the primary afferent fiber in the ipsilateral trigeminal subnucleus caudalis (Figure 3B). Furthermore, AT2R colocalized with CGRP-positive neurons in the trigeminal ganglion (Figure 3C).

### 3.4. Role of Glial Cells in Ang II-Induced Mechanical Allodynia

To determine the role of glial cells in Ang II-induced mechanical allodynia, changes in air puff thresholds were measured after the blockade of glial cell activity (Figure 4). Pretreatment with minocycline (50 or 100 μg), a microglial inhibitor, did not affect air-puff thresholds (Figure 4A). However, intracisternal pretreatment with LAA (8 μg), an astrocyte inhibitor, significantly inhibited Ang II-induced mechanical allodynia (F_(2,18)_ = 3313.3, *p* < 0.05, Figure 4B).

### 3.5. Effects of BTX-A on Ang II-Induced Mechanical Allodynia and Thermal Hyperalgesia

Figure 5 presents the changes in Ang II-induced mechanical allodynia and thermal hyperalgesia after subcutaneous pretreatment with a three-unit (U) dose of BTX-A. Pretreatment with BTX-A did not affect Ang II-induced mechanical allodynia (Figure 5A). However, BTX-A significantly decreased Ang II-induced thermal hyperalgesia (F_(1,12)_ = 1806.24, *p* < 0.05, Figure 5B). 

## 4. Discussion

The present study showed that intracisternal Ang II injection induced mechanical allodynia and thermal hyperalgesia in the orofacial area. Pretreatment with losartan, an AT1R antagonist, blocked only mechanical allodynia, whereas pretreatment with PD123319, an AT2R antagonist, blocked only thermal hyperalgesia. Double immunofluorescence staining revealed that AT1R was localized in astrocytes of the trigeminal subnucleus caudalis and AT2R was localized to the primary afferent fibers. Intracisternal pretreatment with LAA, an astrocyte inhibitor, but not minocycline, a microglial inhibitor, blocked Ang II-induced mechanical allodynia. Furthermore, pretreatment with BTX-A blocked Ang II-induced thermal hyperalgesia but not mechanical allodynia. These results indicate that intracisternally injected Ang II-induced mechanical allodynia and thermal hyperalgesia are differentially mediated by AT1R and AT2R.

It is well-known that Ang II is converted from Ang I by ACE. The physiological function of Ang II is to regulate blood pressure and volume, including vasoconstriction, sodium retention, thirst, and aldosterone synthesis [27]. Recent studies found that intrathecal Ang II injection induced nociceptive scratching behavior in mice [8,28], and that intrathecal treatment with losartan significantly inhibited formalin-induced nociceptive scratching behavior in these animals [9]. These findings suggested that Ang II also contributed to pain processing in the central nervous system. However, systemic circulating Ang II cannot reach the central nervous system as the brain is isolated from circulation by the blood–brain barrier, which consists of specialized endothelium and astrocytes. Therefore, Ang II must be locally synthesized in the central nervous system to activate receptors expressed in this region. The association between Ang II and pain transmission in the central nervous system is supported by previous findings. After formalin injection, the fluorescence intensity of Ang II significantly increases in the ipsilateral superficial dorsal horn (laminae I and II) of the spinal cord [9]. These findings suggest that spinal cord Ang II could function as a neuromodulator in the processing of nociceptive information [9,28]. In the present study, the intracisternal administration of Ang II induced significant nociceptive behaviors, including mechanical allodynia and thermal hyperalgesia. These results, together with previous findings, suggest that Ang II plays a pivotal role in nociceptive information processing in the trigeminal spinal nucleus. Furthermore, Ang II protein expression has been validated in the trigeminal ganglia of rats and humans [13], further supporting the role of Ang II in pain processing in the orofacial area.

Ang II functions via G-protein-coupled receptors AT1R and AT2R [29]. In general, the effects of AT1R activation are offset by the downstream effects of AT2R [30]. Thus, AT2R plays a role in maintaining the balance between the effects of AT1R. The present study showed that intracisternal treatment with losartan, an AT1R antagonist, blocked only mechanical allodynia, whereas intracisternal treatment with PD123319, an AT2R antagonist, blocked only thermal hyperalgesia. These results suggest that mechanical allodynia and thermal hyperalgesia induced by intracisternal Ang II injection are differentially regulated by AT1R and AT2R. These results are also consistent with previous findings that scratching behavior induced by the intrathecal administration of Ang II was mediated by AT1R, but not AT2R [8]. Furthermore, the intrathecal administration of losartan exerted significant antinociceptive effects in the formalin-induced pain test [9]. Previous studies have also shown the involvement of AT2R in pain processing. Several clinical studies demonstrated that a highly selective AT2R antagonist attenuated neuropathic pain [10,31]. However, a recent clinical study failed to identify statistically significant antinociceptive effects of AT2R antagonists on neuropathic pain due to conflicting clinical data [32]. Therefore, the role of AT2R in nociception remains controversial.

Immunofluorescence staining data indicate that AT1R and AT2R play a role in nociceptive processing. AT1R colocalized with the astrocyte marker (GFAP). These results suggest that glial cells contribute to Ang II-induced pronociception. Moreover, intracisternal pretreatment with LAA, an astrocyte inhibitor, but not minocycline, a microglial inhibitor, significantly attenuated mechanical allodynia induced by intracisternal Ang II administration. These behavioral data confirmed the role of astrocyte AT1R in Ang II-induced mechanical allodynia. These results are consistent with those of previous studies. A previous study identified AT1R expression in neurons and astrocytes of mouse spinal cord [9,28]. Furthermore, the intrathecal injection of losartan inhibited tactile allodynia in a streptozotocin-induced diabetic neuropathic pain model [33]. These results, together with the present findings, suggest that mechanical allodynia induced by intracisternal Ang II injection is mediated by AT1R expressed in astrocytes. Further studies are warranted to investigate the underlying cellular and molecular mechanisms.

Immunofluorescence staining also showed that AT2R colocalized with CGRP in the trigeminal subnucleus caudalis and with CGRP-positive neurons in the trigeminal root ganglion. In addition, the intracisternal administration of the AT2R antagonist attenuated thermal hyperalgesia but not mechanical allodynia. These results indicate that AT2R expressed in the primary afferent terminal mediates thermal hyperalgesia induced by intracisternal Ang II injection. These results are consistent with previous findings suggesting that blocking AT2R attenuated neuropathic pain and inflammatory pain [34,35,36,37]. AT2R is expressed in the sensory neurons of peripheral nerve fibers and the dorsal root ganglion in human tissues [38] and expressed in the lumbar dorsal root ganglia in rats [34,36,39]. Immunofluorescence studies have also demonstrated that, in rats, AT2R colocalizes with small-to-medium-substance, P-positive sensory neurons in the lumbar dorsal root ganglion [35,39]. Moreover, previous studies have shown that a highly selective AT2 receptor antagonist attenuates neuropathic pain [10,31] and chronic inflammatory pain [36,40]. The present study demonstrated that AT2R contributes to pain processing using further behavioral data after BTX-A injection. Pretreatment with BTX-A significantly inhibited Ang II-induced thermal hyperalgesia but did not affect Ang II-induced mechanical allodynia. Therefore, AT2R regulates Ang II-mediated thermal hyperalgesia by prompting the release of CGRP and substance P in the terminals of primary afferent fibers.

The present study demonstrated that AT1R was highly expressed on astrocytes, whereas AT2R was expressed on the CGRP of the trigeminal subnucleus caudalis. Mechanical allodynia was induced by intracisternal Ang II injection. Intracisternally injected Ang II acts on synapses in the trigeminal subnucleus caudalis. These experimental results suggest that Ang II, when injected intracisternally, excites astrocytes through AT1R and subsequently induces mechanical allodynia. However, it remains unclear how astrocytes contribute to the induction of mechanical allodynia mediated by large-diameter nerve fibers at the spinal cord level. Therefore, further studies are warranted to elucidate the underlying mechanisms by which intracisternal Ang II administration induces mechanical allodynia. To date, many studies have been conducted regarding the use Ang II-related drugs as clinical treatments. However, only a few experiments have reached the clinical stage. Furthermore, only little evidence from both clinical and preclinical studies suggests that Ang II-related drugs are involved in neuropathic pain. A recent study reported that EMA401, a highly selective AT2R antagonist, exhibited analgesic efficacy in patients with postherpetic neuralgia and painful diabetic neuropathy in clinical studies [10,11]. These results suggest that EMA401 is the first AT2R antagonist analgesic to reach clinical trials. The present study examined the effects of central Ang II on orofacial pain transmission as the blood–brain barrier prevents Ang II in the bloodstream from crossing the central nervous system. These experimental results imply that the RAS exists in the central nervous system, which is distinct from the peripheral nervous system. A previous study reported that the expressions of Ang II and ACE were increased in the spinal cord in a streptozotocin-induced diabetic neuropathic pain model [33]. The present study showed that intracisternal Ang II injection induced severe mechanical allodynia and heat hyperalgesia. Mechanical allodynia is mediated by AT2R on the astroglial cells. In addition to previous studies, the present study demonstrates that RAS, which exists in the central nervous system, is likely to be significantly involved in mechanical allodynia occurring after nerve damage. Therefore, Ang II regulation in the central nervous system shows potential as a new therapeutic drug for the treatment of chronic neuropathic pain.

In addition to the spinal cord, Ang II increases pain sensitivity in the caudal ventrolateral medulla [41] and decreases pain sensitivity in the rostral ventromedial medulla and periaqueductal gray and [42,43,44]. Although these results indicate that Ang II contributes to pain modulation at the supraspinal level in the central nervous system, further studies are required to identify the underlying mechanisms.

In summary, the intracisternal administration of Ang II induced both mechanical allodynia and thermal hyperalgesia. Ang II-induced mechanical allodynia was mediated by AT1R expressed in astrocytes. On the orther hand, AT2R in the primary afferent terminal regulated Ang II-induced thermal hyperalgesia by modulating the release of neuropeptides such as substance P in the terminals of primary afferent fibers. These results suggest that specific types of central Ang II-induced nociception are differentially regulated by AT1R and AT2R. Therefore, specific receptors must be targeted to overcome different types of pain symptoms, as multiple underlying mechanisms contribute to the development of chronic pain.

## Figures and Tables

**Figure 1 biomedicines-11-03279-f001:**
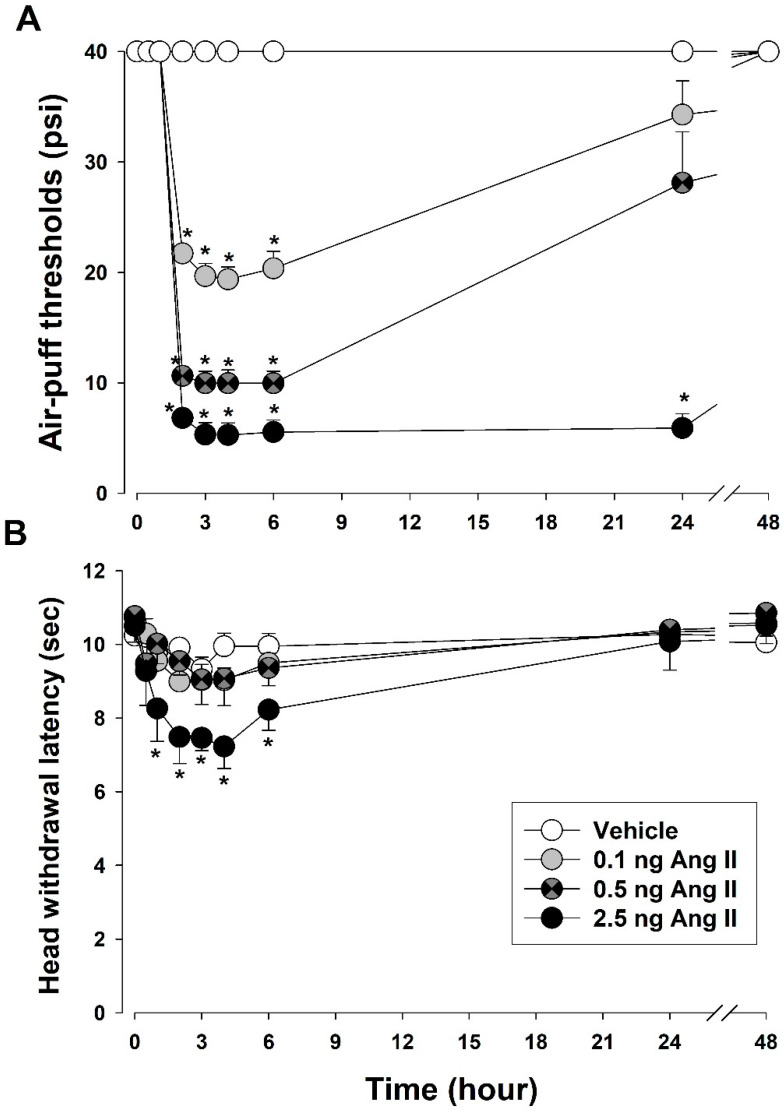
Changes in air-puff thresholds and head-withdrawal latency after intracisternal Ang II injection. (**A**) Intracisternally-injected Ang II (0.1, 0.5, or 2.5 ng) significantly decreased air puff thresholds. Intracisternal vehicle injection did not affect air puff threshold. (**B**) Intracisternal injection of high-dose Ang II (2.5 ng) also significantly decreased head withdrawal latency. Neither low-dose Ang II nor vehicle affected head withdrawal latency. * *p* < 0.05, vehicle vs. Ang II-treated group, *n* = 7 animals/group.

**Figure 2 biomedicines-11-03279-f002:**
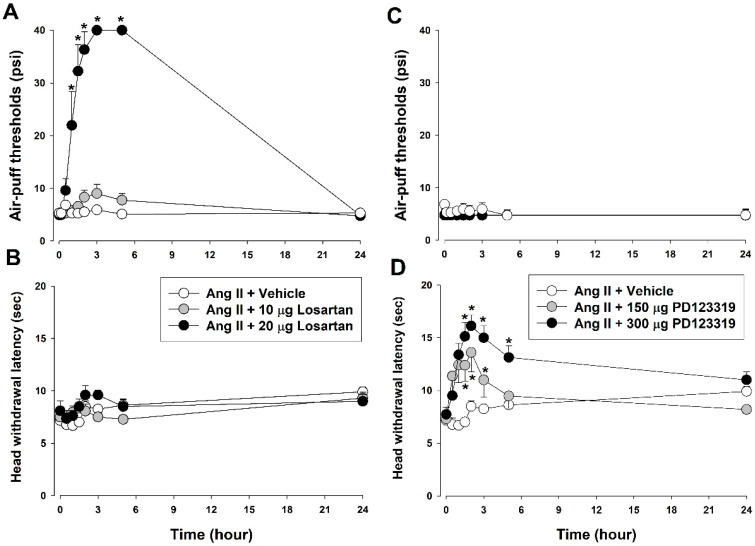
Effects of angiotensin II receptor antagonists on intracisternally injected Ang II-induced mechanical allodynia and thermal hyperalgesia. (**A**) Intracisternal injection of AT1R antagonist losartan (20 μg) blocked Ang II-induced mechanical allodynia. Neither low-dose losartan nor vehicle affected air-puff thresholds. (**B**) Losartan injection did not affect thermal hyperalgesia. (**C**) Intracisternal administration of AT2R antagonist PD123319 did not affect Ang II-induced mechanical allodynia. (**D**) Both doses of PD123319 (150 and 300 μg) significantly decreased thermal hyperalgesia compared with vehicle. * *p* < 0.05, vehicle vs. AT1/2 receptor-antagonist-treated group, *n* = 7 animals/group.

**Figure 3 biomedicines-11-03279-f003:**
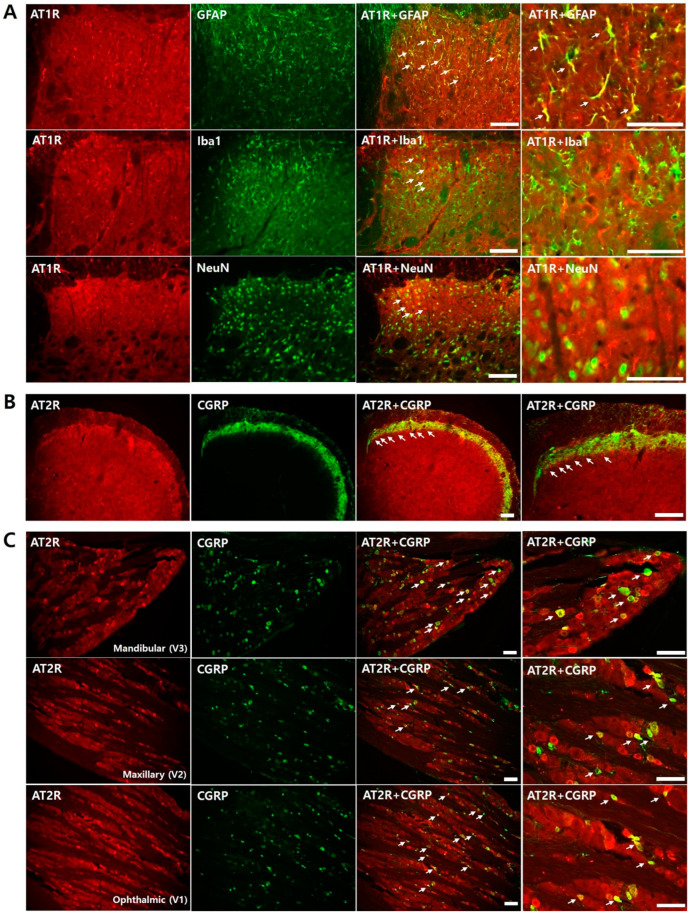
Characterization of AT1R and AT2R in the ipsilateral trigeminal subnucleus caudalis and trigeminal ganglion. (**A**) Double immunofluorescence analysis for AT1R (red) with astrocyte marker (GFAP), microglial marker (Iba1), and neuronal marker (NeuN). AT1R immunoreactive cells were found to mainly be colocalized with GFAP. (**B**) Double immunofluorescence analysis for AT2R (red) colocalized with c-terminal marker (CGRP). AT2R colocalized with CGRP in the ipsilateral trigeminal subnucleus caudalis. (**C**) Double immunofluorescence staining for AT2R (red) with CGRP-positive neurons in the trigeminal ganglion. AT2R colocalized with CGRP-positive neurons in the trigeminal ganglion. Scale bar, 50 μm.

**Figure 4 biomedicines-11-03279-f004:**
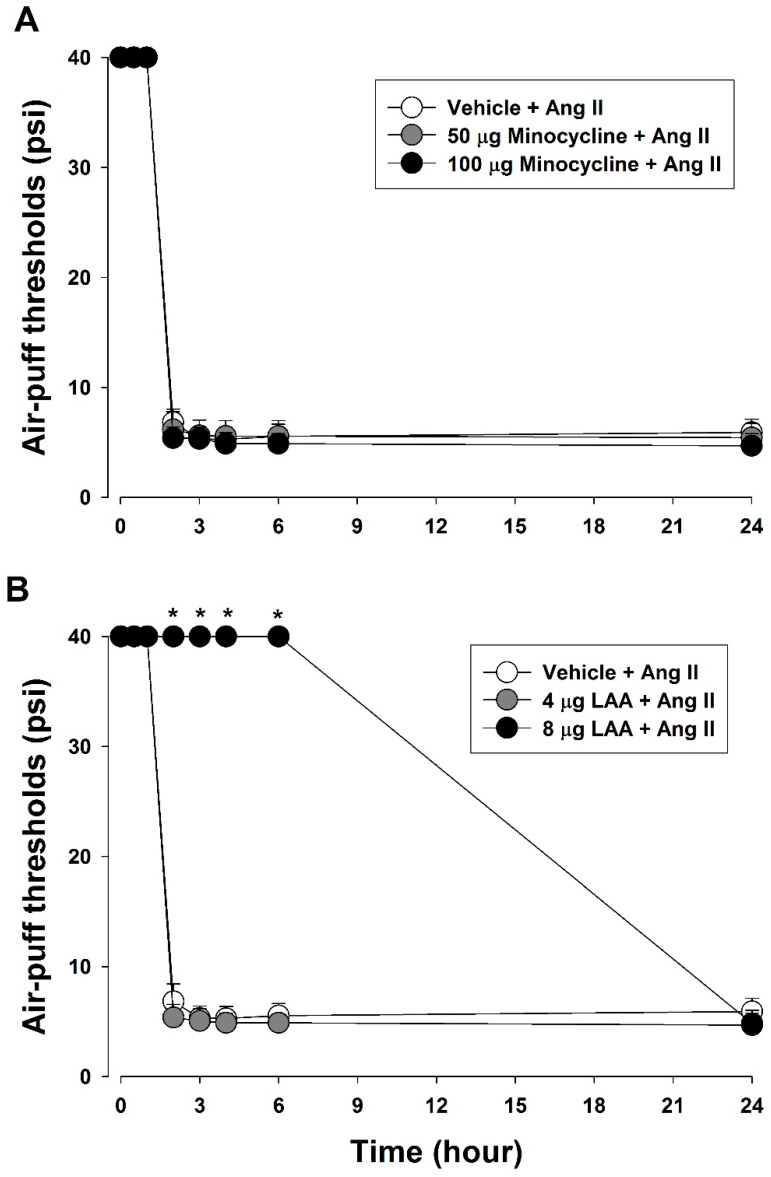
Effects of glial cell inhibition on Ang II-induced mechanical allodynia. (**A**) Pretreatment with minocycline (50, 100 μg), a microglial inhibitor, did not affect air puff thresholds. (**B**) Intracisternal pretreatment with LAA (8 μg), an astrocyte inhibitor, significantly inhibited Ang II-induced mechanical allodynia. * *p* < 0.05, vehicle vs. LAA-treated group, *n* = 7 animals/group.

**Figure 5 biomedicines-11-03279-f005:**
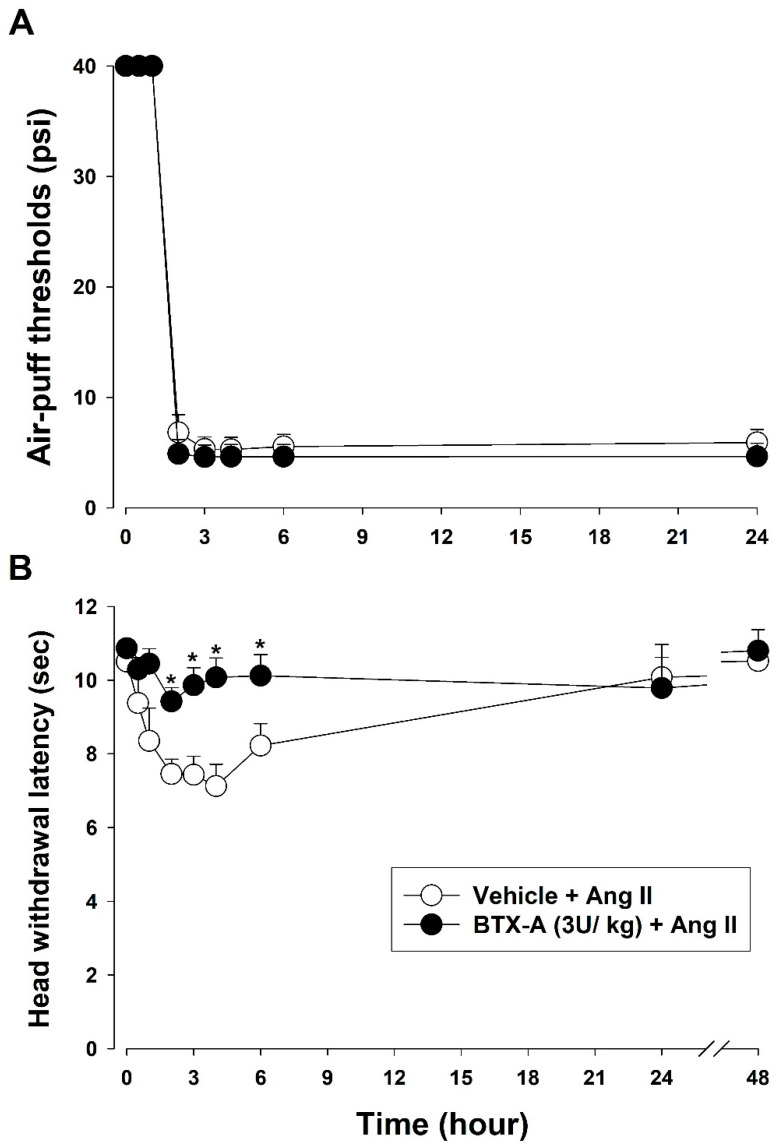
Effects of botulinum toxin type A on Ang II-induced mechanical allodynia and thermal hyperalgesia. (**A**) Pretreatment with BTX-A did not affect Ang II-induced mechanical allodynia. (**B**) Pretreatment with BTX-A significantly inhibited Ang II-induced thermal hyperalgesia. * *p* < 0.05, vehicle vs. BTX-A-treated group, *n* = 7 animals/group.

## Data Availability

The data presented in this study are available on request from the corresponding author (dkahn@knu.ac.kr).

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
