# Peer review of "Differential Regulation of Intracisternally Injected Angiotensin II-Induced Mechanical Allodynia and Thermal Hyperalgesia in Rats"

_biomedicines, 2023, doi:10.3390/biomedicines11123279_

Round 1

Reviewer 1 Report

Comments and Suggestions for Authors

The authors Park et al. have reported the mechanism underlying differential involvement of the angiotensin I and II receptors in the trigeminal spinal subnucleus caudalis (Vc) and trigeminal ganglion in mechanical allodynia and heat hyperalgesia in the face. After the intracisternal administration of angiotensin II, mechanical allodynia and heat hyperalgesia occur in the face, and intracisternal administration of AT1 receptor antagonist caused blockade of the mechanical allodynia, and AT2 receptor antagonist administration caused heat hyperalgesia. And also, the AT1 receptor was co-expressed in GFAP immuno-positive cells (astrocytes), and the AT2 receptor was in CGRP-positive cells. The authors concluded that the AT1 receptor is involved in orofacial mechanical allodynia via the astrocyte mechanism, whereas the AT2 receptor is involved in orofacial heat hyperalgesia via the neuropeptide mechanism. The results are fascinating. However, several points should be clarified. The followings are comments on the current manuscript:

Major

1.       The authors addressed that the AT1 receptor antagonist administration suppressed mechanical allodynia caused by the angiotensin II intracisternal administration via astroglia activation. Are the AT2 and AT1 receptors expressed in astroglia? The authors need to show how the astroglia is activated by angiotensin II. Large myelinated fibers are thought to be involved in mechanical allodynia. How do the astroglial cells contribute to modulating large ganglion neurons?

2.       Does the intracisternal administration of angiotensin I cause astroglial cell activation and mechanical allodynia but not heat hyperalgesia?

3.       Photomicrographs of AT1 and AT2 receptors in Fig. 3 are not of enough quality to address if those stainings are positive. Double immunohistochemical stainings are also challenging to define.

4.       The quantitative analysis of AT1R + GFAP and AT2R + CGRP immunopositive cells should be conducted in Vc. Photomicrographs of the contralateral sections should be included in Fig.3. Positive cells should be pointed by the arrows or arrowhead.

Minor

1.       Please provide the high-magnification photomicrographs in each section in Fig. 3.

2.       P4 line 179: “withdrawal latency time” should be “withdrawal latency”, and also those in Figs. 1, 2, and 5.

3.       P4 line 184: “Attenuation of air puff threshold” should be “Decrement of air puff threshold”

4.       Fig. 2: “Effects of angiotensin II” shold be “Effects of angiotensin I and II”

5.       P9 line 265: “ was localized to “ should be  “ was localized in “

6.       P11 line 348: delete “and”

7.       P11 line 355: “ substance P” should be “ CGRP and substance P”

Comments on the Quality of English Language

N/A

Author Response

Dear Reviewer

It is my pleasure to re-submit my results to your top classed journal, "Biomedicines". The previous manuscript reference number is Biomedicines-2395685. Please find enclosed the article titled “Differential regulation of intracisternally injected angiotensin II-induced mechanical allodynia and thermal hyperalgesia in rats” for consideration of publication in "Biomedicines".

We corrected the manuscript and inserted new data in response to the reviewers’ comments. The lists are at the end of the section. As detailed below, we have addressed all of the comments and concerns raised by the reviewers. We thank the reviewers for their insightful comments, and feel that the manuscript is now more improved as a result of the revisions.

I look forward to your response.

Should you have any questions, please feel free to contact me.

Thank you

Yours sincerely

Dong-Kuk Ahn DDS, PhD

Reviewer 2 Report

Comments and Suggestions for Authors

Make some link (connection) to clinical situations in the introduction section. Are angiotensin-acting therapeutical interventions currently already available in the clinic, or refer to previous findings?

Line 76: put a comma after the first and. Perhaps provide a drawning of the positioning of the tube and fixation point (or clinical picture). 

In the discussion section the authors should elaborate more on the possible clinical implications of their findings. Currently, their conclusions remain very high level and without any practical details. For which types of chronic pain syndromes will these findings possibly be useful in the future?

Author Response

Dear Editor

It is my pleasure to re-submit my results to your top classed journal, "Biomedicines". The previous manuscript reference number is Biomedicines-2395685. Please find enclosed the article titled “Differential regulation of intracisternally injected angiotensin II-induced mechanical allodynia and thermal hyperalgesia in rats” for consideration of publication in "Biomedicines".

We corrected the manuscript and inserted new data in response to the reviewers’ comments. The lists are at the end of the section. As detailed below, we have addressed all of the comments and concerns raised by the reviewers. We thank the reviewers for their insightful comments, and feel that the manuscript is now more improved as a result of the revisions.

I look forward to your response.

Should you have any questions, please feel free to contact me.

Thank you

Yours sincerely

Dong-Kuk Ahn DDS, PhD

Round 2

Reviewer 1 Report

Comments and Suggestions for Authors

The revised manuscript is fine to me. I do not have any concerns about accepting this paper.